# A preliminary cost-utility analysis of routine myasthenia gravis and thyroid dysfunction screening in acquired comitant Esotropia

**Worapot Srimanan** *, **Phawasutthi Keokajee**◎, **Sunita Sawangsribanterng**◎

Division of Ophthalmology, Phramongkutklao Hospital, Bangkok, Thailand

◎ These authors contributed equally to this work
* drworapotsmn@gmail.com

## Abstract

This study evaluated the cost-effectiveness of routine screening for acetylcholine receptor antibody (AChR-Ab) and thyroid function tests (TFT) in patients with acquired comitant esotropia (ACE) without overt signs of myasthenia gravis (MG) or thyroid eye disease (TED). A retrospective cost–utility analysis was conducted in 110 patients at a Thai tertiary hospital between 2014 and 2024. A decision tree combined with a 10-year Markov model compared two strategies: no routine screening (symptom-triggered testing) and universal baseline screening with AChR-Ab and TFT. Costs were expressed in 2024 Thai Baht (THB) from a healthcare sector perspective, and outcomes were measured in quality-adjusted life years (QALYs). Model uncertainty was assessed using one-way sensitivity analyses and probabilistic sensitivity analysis with 10,000 simulations, incorporating downstream costs of follow-up, confirmatory evaluation, and treatment. In the base-case analysis incorporating real-world diagnostic accuracy, universal screening yielded higher QALYs (109.56 vs. 105.45) but also higher costs (฿2,826,680 vs. ฿1,653,500), resulting in an incremental cost-effectiveness ratio (ICER) of ฿285,360 per QALY gained. This exceeded Thailand's willingness-to-pay threshold of ฿160,000–200,000 per QALY, indicating that universal screening was not cost-effective. Probabilistic sensitivity analysis showed that most simulations were located in the northeast quadrant of the cost-effectiveness plane, reflecting greater effectiveness with higher cost, with many exceeding the willingness-to-pay threshold. Key drivers included MG prevalence, utility loss from undiagnosed MG, and AChR-Ab test cost. TFT screening contributed minimal benefit due to the very low prevalence of thyroid dysfunction. Universal AChR-Ab screening may improve early detection of MG in ACE, but it was not cost-effective under current assumptions. Exploratory targeted screening appeared relatively more efficient, and symptom-triggered thyroid testing may be a more appropriate approach. These findings are preliminary and require validation in larger studies.

**Data availability statement:** All relevant data are within the manuscript and its Supporting Information files.

**Funding:** The author(s) received no specific funding for this work.

**Competing interests:** The authors have declared that no competing interests exist.

## Introduction

Acquired comitant esotropia (ACE) is defined by the sudden onset of esodeviation with relatively equal ocular misalignment in all gaze positions [1]. While often classified as idiopathic, ACE may be the initial or sole manifestation of systemic disorders such as myasthenia gravis (MG) or thyroid eye disease (TED), even in the absence of hallmark signs such as ptosis, proptosis, or lid retraction [2].

Emerging evidence shows that ocular MG can present exclusively as comitant esotropia [3,4], while TED—traditionally associated with lid changes and orbital soft-tissue signs—may initially appear as isolated comitant esotropia or divergence insufficiency [5]. These atypical presentations can delay recognition of the underlying systemic disease, resulting in missed opportunities for early intervention and potentially prolonged morbidity.

Baseline laboratory screening, particularly acetylcholine receptor antibody (AChR-Ab) assays and thyroid function tests (TFT), offers the potential for earlier detection of these treatable conditions. AChR-Ab testing for ocular MG has demonstrated sensitivities of 70–80% and specificity of approximately 98–99% [6,7]. TFTs for hypothyroidism show sensitivities and specificities around 90% and 92%, respectively [8–10]. Nevertheless, imperfect sensitivity inevitably results in false negatives, leading to undiagnosed disease, reduced quality of life, and higher downstream healthcare costs.

The cost-effectiveness of routine screening in ACE is not well established. In resource-limited healthcare systems, such as Thailand, the financial burden of universal testing must be balanced against the potential gains in quality-adjusted life years (QALYs). Previous Thai work, including the study by Lekskul et al. [11], has described the clinical characteristics of ACE but has not explored the economic impact of screening for MG or TED. This gap limits evidence-based recommendations for diagnostic pathways in this population.

To address this gap, we conducted a preliminary cost-utility analysis of routine AChR-Ab and TFT screening in patients with ACE, incorporating real-world diagnostic performance data. Using a decision tree combined with a 10-year Markov model, we compared universal baseline screening with no routine screening (symptom-triggered testing), in which diagnostic evaluation was performed only when clinical suspicion arose during follow-up, evaluating both clinical outcomes and economic consequences across true-positive, false-negative, false-positive, and true-negative scenarios. Given the low expected prevalence of underlying systemic disease in ACE, economic modeling based on single-center data may be sensitive to small variations in observed event rates, underscoring the exploratory nature of this analysis.

## Materials and methods

### Study design and setting

This retrospective cost–utility analysis evaluated 110 consecutive patients diagnosed with ACE at a tertiary care ophthalmology clinic in Thailand between January 2014 and May 2024. Researchers accessed data during September 2025. Data for

individual participants was only accessed during data collection. The study protocol was approved by the local Institutional Review Board (approval number S054h/68_Exp), and all procedures adhered to the principles of the Declaration of Helsinki.

### Eligibility criteria

Inclusion criteria comprised patients of all ages with documented clinical and ophthalmologic assessments consistent with ACE, comitant esodeviation in all gaze directions and no classic signs of MG or TED, such as ptosis, proptosis, or lid retraction. Patients with prior diagnoses of MG or TED, or those with incomplete clinical or laboratory records, were excluded.

### Diagnostic strategies compared

Two diagnostic strategies were modeled:

1. **No Routine Screening (Symptom-Triggered Testing)** – Patients did not undergo baseline laboratory testing. Diagnostic evaluation for MG or thyroid dysfunction was performed only if clinical suspicion arose during follow-up based on emerging signs or symptoms.

2. **Universal Screening** – All patients underwent baseline AChR-Ab testing and TFT, including thyroid-stimulating hormone (TSH), free triiodothyronine (FT3), and free thyroxine (FT4).

### Targeted screening strategy

In addition to the two primary strategies, we modeled a third targeted screening approach designed to reflect a pragmatic compromise in Thai tertiary practice. In this strategy, only patients with higher pre-test probability of systemic disease—defined as those presenting to referral centers with atypical features (e.g., adult-onset diplopia, rapid symptom onset, or variable deviation)—underwent baseline AChR-Ab testing. The proportion of patients meeting these criteria was conservatively set at 30% of the ACE cohort, consistent with published referral patterns [2,11–14]. Test performance and downstream utilities were modeled as in the universal screening arm. Patients not meeting targeted criteria were managed under the no-screening pathway. Costs and QALYs were estimated over the same 10-year horizon with 3% annual discounting.

This targeted subgroup proportion (30%) was based on assumed referral patterns rather than empirically derived from the study cohort. In the current model, we did not assign a different disease prevalence to the targeted subgroup, nor did we model a separate targeting test with defined sensitivity or specificity. Instead, the targeted strategy was implemented as a proportional application of the universal screening pathway to a predefined subset of patients, with identical assumptions about diagnostic accuracy, cost, and utility.

Because no validated risk-stratification rule was available, the targeted screening strategy was modeled as an exploratory scenario rather than a formally validated diagnostic comparator. This approach was intended to approximate real-world clinical behavior in tertiary-care settings, where testing is often selectively applied based on clinical suspicion, rather than to represent a calibrated prediction model with defined sensitivity or specificity. Accordingly, this strategy should be interpreted as a proportional reduction in testing volume rather than a formally risk-enriched screening strategy, and was included primarily as an exploratory scenario analysis.

### Diagnostic accuracy assumptions

Real-world sensitivity and specificity values were incorporated into the universal screening arm to reflect clinical performance:

- **AChR-Ab testing**: Sensitivity = 75%, specificity = 98%, based on large cohort data for ocular MG (70.9% sensitivity in one study [6] and ~80% sensitivity approaching 99% specificity in a tertiary care setting [7]).

- **TFT**: TSH-first reflex FT4/FT3 approach, sensitivity = 90%, specificity = 92%, derived from diagnostic accuracy studies in referred populations and conservatively adjusted for a screening context [8–10].

These parameters determined the probability of true-positive (TP), false-negative (FN), false-positive (FP), and true-negative (TN) outcomes for each test in the decision tree model.

The decision tree (S1 Fig) modeled all 110 patients in the screening arm undergoing both AChR-Ab and TFT, with diagnostic outcomes determined by observed prevalence (OMG = 2.7%, hypothyroidism = 0.9%) and published accuracy data [6–10]. True positives (TP) received early treatment with corresponding utility gains (OMG = 0.872; hypothyroidism = 0.94), false negatives (FN) experienced delayed diagnosis with lower utility (OMG = 0.739), false positives (FP) incurred additional confirmatory investigations and follow-up visits, resulting in associated short-term medical costs without long-term disutility, and true negatives (TN) retained full health utility (1.00). In the no-screening strategy, delayed diagnosis for MG cases reduced long-term utility. Each diagnostic pathway in the decision tree was explicitly assigned corresponding downstream costs and utilities to reflect real-world clinical management.

## Cost inputs

All costs were calculated in 2024 Thai Baht (THB). In our institution, MRI of the brain and orbit with gadolinium is routinely performed for all ACE patients at baseline to exclude structural or neurological causes, regardless of MG/TED suspicion; therefore, this cost was applied equally to both strategies.

The model incorporated both initial and downstream direct medical costs across all diagnostic pathways. These included baseline diagnostic tests (AChR-Ab, thyroid function tests), follow-up and monitoring costs (outpatient visits), confirmatory investigations (e.g., chest CT for thymoma evaluation), and disease-specific treatments (e.g., pyridostigmine for myasthenia gravis and levothyroxine for hypothyroidism).

Representative unit costs included AChR-Ab testing (฿1,900), thyroid function tests (TSH ฿170, FT3 ฿170, FT4 ฿150), chest CT (฿8,200), MRI brain/orbit (฿16,000), and outpatient follow-up visits (฿750 per visit). We assumed two follow-up visits per year over three years (six visits per patient), consistent with institutional practice.

These costs were assigned to relevant decision tree branches according to diagnostic outcomes (true positive, false negative, false positive, and true negative). Detailed cost inputs, including unit costs, quantities, and their application within each pathway, are provided in S3 Table.

## Utility inputs and outcome modeling

Based on test accuracy and disease prevalence, the model incorporated the probability distribution of diagnostic outcomes (true positive, false negative, false positive, and true negative). Each health state was assigned a utility value from published literature:

- Early diagnosed ocular MG: 0.872

- Subclinical hypothyroidism (treated): 0.94

- Idiopathic ACE (true negative): 1.00

- Delayed-diagnosed MG: 0.739

- Delayed-diagnosed hypothyroidism: 0.94

- Missed underlying condition (treated as idiopathic): 0.965

The decision tree simulated the detection and management of MG and hypothyroidism under each strategy. Long-term outcomes were projected using a Markov model with annual cycles over a 10-year horizon. The Markov model applied half-cycle correction, and "Death" was included only to represent background age-related mortality rather than ACE-specific fatality. A schematic of the model structure is provided in S2 Fig. Both costs and utilities were discounted at 3% annually, consistent with Thai Health Technology Assessment (HTA) guidelines.

The Markov model was designed as a simplified state-transition framework and did not explicitly model all intermediate clinical transitions, including relapse frequency, treatment-related adverse effects, or detailed progression from ocular to generalized myasthenia gravis. Health-state utilities were applied as average state-based values over the analytic horizon rather than as fully time-varying trajectories, and therefore may not capture short-term fluctuations in symptoms or treatment response.

The primary outcome was the incremental cost-effectiveness ratio (ICER), expressed as cost per QALY gained.

Absolute cohort-level QALY totals represent population-average outcomes over the 10-year horizon and should not be interpreted as disease-specific utility estimates for affected patients alone.

### Sensitivity analyses

Model robustness was evaluated using:

1. One-Way Sensitivity Analysis – Key parameters (MG prevalence, hypothyroidism prevalence, AChR-Ab and TFT accuracy, test costs, and utility values) were varied by ±20% from base-case estimates. Utility parameters were included in one-way sensitivity analyses; however, alternative structural assumptions regarding utility trajectories over time were not evaluated. Results were expressed as changes in net monetary benefit (NMB) at a WTP of ฿160,000/QALY and displayed in a Tornado Diagram.

2. Probabilistic Sensitivity Analysis (PSA) – A total of 10,000 Monte Carlo simulations were performed.

   - Probability and utility parameters were assigned beta distributions.

   - Cost parameters were assigned gamma distributions. Each simulation generated an ICER, plotted on a cost-effectiveness plane and summarized as a distribution histogram.

Cost-effectiveness was assessed relative to Thai GDP-based WTP thresholds of ฿160,000 and ฿200,000 per QALY gained [15,16]. Unless otherwise specified, all costs and QALYs are presented as cohort totals for 110 patients; however, ICER values are scale-invariant and thus directly interpretable on a per-patient basis.

### Ethics approval and consent to participate

The Institutional Review Board of the Royal Thai Army Medical Department reviewed and approved the study protocol (approval number S054h/68_Exp). The requirement for written informed consent was waived by the Institutional Review Board due to the study's retrospective design. Participant data were kept anonymous and confidential.

## Results

### Demographic and clinical characteristics

A total of 110 patients diagnosed with ACE between January 2014 and May 2024 were included (Table 1). The mean age at presentation was 30.3±20.5 years (median 28; range 3–84 years), with an equal sex distribution (55 females, 50%). The majority of patients presented with Type 2 (n=51, 46.4%) or Type 3 (n=51, 46.4%) ACE, while Type 1 accounted for only 7.2% (n=8). The mean spherical equivalent refractive error was −1.90±3.52 diopters (median −0.50 D; range −24.00 to +3.00 D).

**Table 1. Demographic and Clinical Characteristics of Patients with ACE (N = 110).**

| Variable | Summary |
|---|---|
| Sex | Female: 55 (50%), Male: 55 (50%) |
| Type of ACE | Type 1: 8 (7.2%), Type 2: 51 (46.4%), Type 3: 51 (46.4%) |
| TFT Result | Normal: 109 (99.1%), Abnormal: 1 (0.9%) |
| AChR-Ab Result | Negative: 107 (97.3%), Positive: 3 (2.7%) |
| Age (years) | Mean: 30.3 ± 20.5; Median: 28.0 (range 3–84) |
| Refractive Error (D) | Mean: −1.90 ± 3.52; Median: −0.50 (range −24.00 to +3.00) |
| Onset Duration (years) | Mean: 2.87 ± 4.38; Median: 1.75 (range 0.01–30.00) |

This table summarizes demographic and clinical characteristics, laboratory findings, and refractive status of ACE patients included in the study.

Abbreviations: ACE = acquired comitant esotropia; TFT = thyroid function test; AChR-Ab = acetylcholine receptor antibody; D = diopters.

Abnormal TFT results were detected in one patient (0.9%), while three patients (2.7%) tested positive for AChR-Ab and were subsequently diagnosed with ocular myasthenia gravis (OMG). Detailed characteristics of the four patients with positive laboratory findings are presented in S1 Table.

## Clinical and diagnostic findings

Among the three OMG patients, AChR-Ab levels ranged from borderline positivity (confirmed by repetitive nerve stimulation) to markedly elevated titers. The single patient with subclinical hypothyroidism had elevated TSH with normal FT3 and FT4 levels. Neuroimaging—MRI brain/orbit or CT chest—was performed in all four cases and excluded thymoma, TED, or other structural lesions. All four patients experienced improvement in ocular alignment or diplopia following targeted therapy: pyridostigmine in OMG cases and levothyroxine in hypothyroidism.

## Utility values

QALY weights used in the cost-utility model were derived from validated literature sources and stratified by disease type and severity (S2 Table). For OMG, early diagnosis and treatment were associated with a utility of 0.872 [17], compared to 0.739 for undiagnosed MG [17]. Subclinical hypothyroidism carried a utility of 0.94, while overt hypothyroidism (treated) was assigned 0.940 [18]. TED utilities ranged widely, depending on severity, from 0.300 for severe cases with constant diplopia and large proptosis to 0.600 for mild cases without diplopia [19]. Because MG and thyroid dysfunction were present in only a small proportion of patients, the overall cohort-level QALY estimates were predominantly influenced by individuals remaining in high-utility health states.

## Cost–utility analysis

**Base-case analysis.** In the primary base-case analysis, universal screening produced 109.56 discounted QALYs per cohort of 110 patients (0.996 QALYs per patient) at a cost of ฿2,826,680, compared with 105.45 QALYs (0.959 QALYs per patient) and ฿1,653,500 for no routine screening (symptom-triggered testing). This corresponded to an incremental gain of 4.11 QALYs at an additional cost of ฿1,173,180. The resulting incremental cost-effectiveness ratio (ICER) was ฿285,360 per QALY gained, which is above Thailand's willingness-to-pay (WTP) threshold of ฿160,000–200,000 per QALY [15,16]. A summary of cohort-level costs and QALYs is shown in Table 2, with a detailed cost breakdown provided in S3 Table.

**Table 2. Cost and QALY Summary by Diagnostic Strategy (10-Year Horizon, 3% Discount; per cohort of 110 patients).**

| Strategy | 10-Year Discounted QALYs (per 110) | Total Cost (THB, per 110) | ICER vs. No routine screening (symptom-triggered testing) (THB/QALY) |
|---|---|---|---|
| Universal screening | 109.56 | 2,826,680 | 285,360 |
| No routine screening (symptom-triggered testing) | 105.45 | 1,653,500 | Reference |

Values represent cohort totals for 110 patients over a 10-year horizon, discounted at 3% annually. Incremental cost-effectiveness ratio (ICER) is calculated as the additional cost per QALY gained with universal screening compared with no routine screening (symptom-triggered testing). Utilities derived from published sources; costs include laboratory testing, neuroimaging, follow-up, and treatment. Per-patient QALY estimates can be derived by dividing cohort totals by 110.

Abbreviations: ACE, acquired comitant esotropia; AChR-Ab, acetylcholine receptor antibody; TFT, thyroid function test; TSH, thyroid-stimulating hormone; FT3, free triiodothyronine; FT4, free thyroxine; MG, myasthenia gravis; OMG, ocular myasthenia gravis; QALY, quality-adjusted life year; ICER, incremental cost-effectiveness ratio; THB, Thai Baht.

Analyses were conducted from the Thai healthcare sector perspective, including direct medical costs related to patient care (e.g., diagnostic testing, follow-up visits, confirmatory investigations, and treatment). Capital and implementation costs—such as laboratory platform acquisition, assay setup, quality-control programs, and personnel training—were not included, as they are not typically included in a healthcare-sector perspective.

**Scenario analysis.** In the scenario analysis, universal screening yielded 933.68 QALYs per cohort (8.49 per patient), compared with 899.48 QALYs (8.18 per patient) under no routine screening. Under these assumptions, universal screening resulted in higher health outcomes at lower costs; however, this scenario reflects optimistic assumptions rather than real-world diagnostic accuracy. These high absolute QALY values reflect the population-average nature of the cohort, in which the majority of patients remained in health states close to full utility, and therefore have limited interpretive value beyond illustrating relative differences between strategies under optimistic assumptions.

**Targeted screening analysis.** To explore a pragmatic middle-ground option, we modeled a targeted screening strategy in which 30% of ACE patients—those presenting with higher pre-test probability of systemic disease (adult-onset diplopia, rapid onset, variable deviation)—underwent baseline AChR-Ab testing. Test performance, utilities, and downstream pathways were modeled identically to the universal screening arm, applied only to the targeted subgroup.

Over the 10-year horizon, targeted screening produced 106.70 discounted QALYs at a total cost of ฿1,986,200, compared with 105.45 QALYs and ฿1,653,500 for no routine screening (symptom-triggered testing). This corresponded to an incremental gain of 1.25 QALYs at an additional cost of ฿332,700, yielding an ICER of approximately ฿266,160 per QALY (Table 3). While this ICER remains above the Thai WTP threshold of ฿160,000–200,000/QALY, it is notably lower than the ICER observed for universal screening (฿285,360/QALY), indicating relatively improved cost-efficiency compared with universal screening. These comparative results for all three strategies are illustrated in Fig 1, which visually demonstrates the relative efficiency of targeted versus universal screening compared with no routine screening (symptom-triggered testing).

Because the targeted subgroup was not modeled with distinct prevalence or selection accuracy, the observed efficiency reflects a proportional reduction in screening costs rather than a true enrichment of disease probability.

**Incremental cost-effectiveness ratio (ICER) calculation.** The ICER was calculated as:

$$ICER = \frac{CostScreen - Cost\ No\ Screen}{QALYs\ Screen - \ QALYs\ No\ Screen}$$

Using the primary base-case estimates over a 10-year horizon with a 3% annual discount:

$$ICER = \frac{2,826,680 - 1,653,500}{109.56 - 105.45} = \ 285,360 \ \textit{THB per QALY gained}$$

**Table 3. Cost and QALY Summary Including Targeted Screening (10-Year Horizon, per cohort of 110 patients, 3% discount).**

| Strategy | 10-Year Discounted QALYs | Total Cost (THB) | ICER vs. No routine screening (THB/QALY) |
|---|---|---|---|
| No routine screening | 105.45 | 1,653,500 | Reference |
| Targeted screening | 106.70 | 1,986,200 | 266,160 |
| Universal screening | 109.56 | 2,826,680 | 285,360 |

Values represent discounted quality-adjusted life years (QALYs) and total costs (Thai Baht, THB) over 10 years, discounted at 3% annually. All values are reported as cohort totals for 110 patients; see Methods for clarification on ICER scaling. Incremental cost-effectiveness ratios (ICERs) are calculated relative to no routine screening (symptom-triggered testing) for each strategy. Utilities were derived from published sources; costs include laboratory testing, neuroimaging, follow-up, and treatment.

Abbreviations: ACE, acquired comitant esotropia; AChR-Ab, acetylcholine receptor antibody; TFT, thyroid function test; TSH, thyroid-stimulating hormone; FT3, free triiodothyronine; FT4, free thyroxine; MG, myasthenia gravis; OMG, ocular myasthenia gravis; QALY, quality-adjusted life year; ICER, incremental cost-effectiveness ratio; THB, Thai Baht..

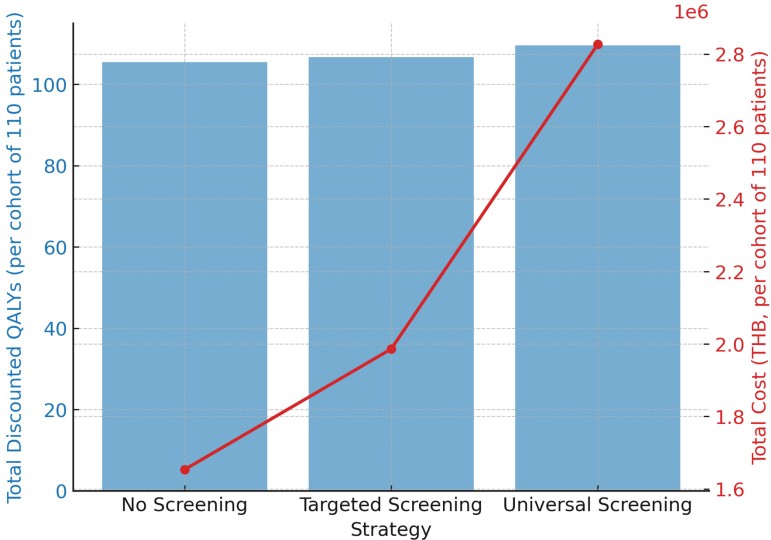

**Fig 1. Cost-Effectiveness Comparison of Diagnostic Strategies (per cohort of 110 patients).** Bar plot comparing total discounted QALYs (blue bars) and total costs (red line) for no routine screening, targeted screening, and universal screening over a 10-year horizon. Targeted screening yielded a lower ICER (฿266,160/QALY) than universal screening (฿285,360/QALY), indicating relatively greater efficiency, although both exceeded Thai willingness-to-pay thresholds. Abbreviations: ACE, acquired comitant esotropia; AChR-Ab, acetylcholine receptor antibody; TFT, thyroid function test; MG, myasthenia gravis; OMG, ocular myasthenia gravis; QALY, quality-adjusted life year; ICER, incremental cost-effectiveness ratio; THB, Thai Baht.

These values reflect per-patient averages scaled from the cohort of 110 patients, ensuring consistency between cost and QALY estimates. Under this specification, universal screening with AChR-Ab and TFT exceeded Thailand's willingness-to-pay threshold of ฿160,000–200,000/QALY [15,16], indicating that while screening improves health outcomes, it is not cost-effective at current cost levels.

These cost differences were driven not only by baseline screening expenses but also by downstream costs associated with follow-up visits, confirmatory investigations, and treatment across diagnostic pathways.

**One-way sensitivity analysis.** One-way sensitivity analysis identified MG prevalence, the utility decrement associated with delayed MG diagnosis, and the unit cost of AChR-Ab testing as the most influential parameters affecting model outcomes. MRI costs, hypothyroidism prevalence, TFT costs, and utilities for diagnosed OMG or idiopathic ACE had only moderate or minor effects. (Fig 2).

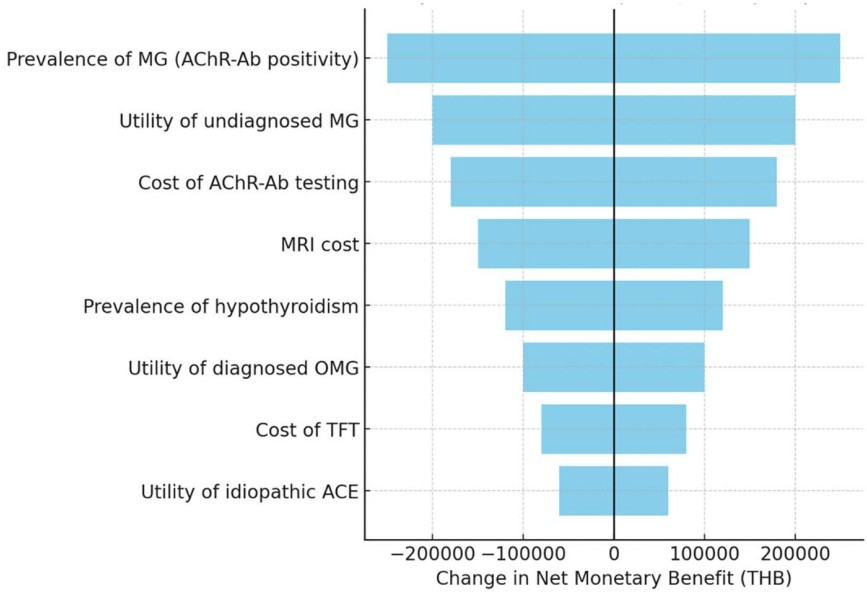

**Fig 2. Tornado Diagram of One-Way Sensitivity Analyses (per cohort of 110 patients).** The tornado diagram shows the relative influence on model outcomes (expressed as net monetary benefit [NMB]) at a willingness-to-pay threshold of ฿160,000 per QALY. The prevalence of MG (AChR-Ab positivity), the utility decrement from undiagnosed MG, and the cost of AChR-Ab testing were the most influential factors, while utilities of idiopathic ACE and diagnosed OMG had smaller effects. Bars indicate ±20% variation from base-case values. Plotted values are based on cohort totals for 110 patients; see Methods for clarification on ICER scaling. Abbreviations: ACE, acquired comitant esotropia; AChR-Ab, acetylcholine receptor antibody; MG, myasthenia gravis; OMG, ocular myasthenia gravis; TFT, thyroid function test; QALY, quality-adjusted life year; NMB, net monetary benefit; THB, Thai Baht.

**Probabilistic sensitivity analysis.** The probabilistic sensitivity analysis (PSA) included 10,000 Monte Carlo simulations that simultaneously varied disease prevalence, test accuracy, costs, and utility values across plausible distributions. The distribution of ICER values was tightly clustered around the base-case estimate of ฿285,360/QALY, with nearly all simulations above Thailand's lower WTP threshold of ฿160,000/QALY (Fig 3). The cost-effectiveness plane (Fig 4) further demonstrated that most simulated outcomes were located in the northeastern quadrant, indicating that screening was more effective but more costly in the majority of scenarios. Together, these findings support the consistency of the unfavorable cost-effectiveness profile of universal screening for ACE.

A cost-effectiveness acceptability curve (CEAC) is provided in S3 Fig, illustrating the probability of universal AChR-Ab screening being cost-effective across a range of willingness-to-pay thresholds. At Thailand's thresholds of ฿160,000–200,000/QALY, the probability of cost-effectiveness remained low, reflecting that screening is unlikely to be economically favorable under current cost structures.

**Price–prevalence thresholds.** A two-way threshold analysis was conducted to identify conditions under which AChR-Ab screening would fall below Thailand's WTP threshold. At the observed prevalence of 2.7%, the test price would need to fall below ฿1,100 for universal screening to become cost-effective at ฿160,000/QALY. Conversely, at the current test price of ฿1,900, MG prevalence would need to exceed 5.0% to achieve cost-effectiveness. The contour plot (Fig 5) illustrates combinations of prevalence and test cost where universal screening is economically favorable, providing a practical benchmark for policy and laboratory pricing negotiations.

## Discussion

In Thailand's resource-conscious healthcare environment, where tertiary centers act as referral hubs for complex neuro-ophthalmic presentations, diagnostic strategies must balance clinical thoroughness with economic value. Acquired

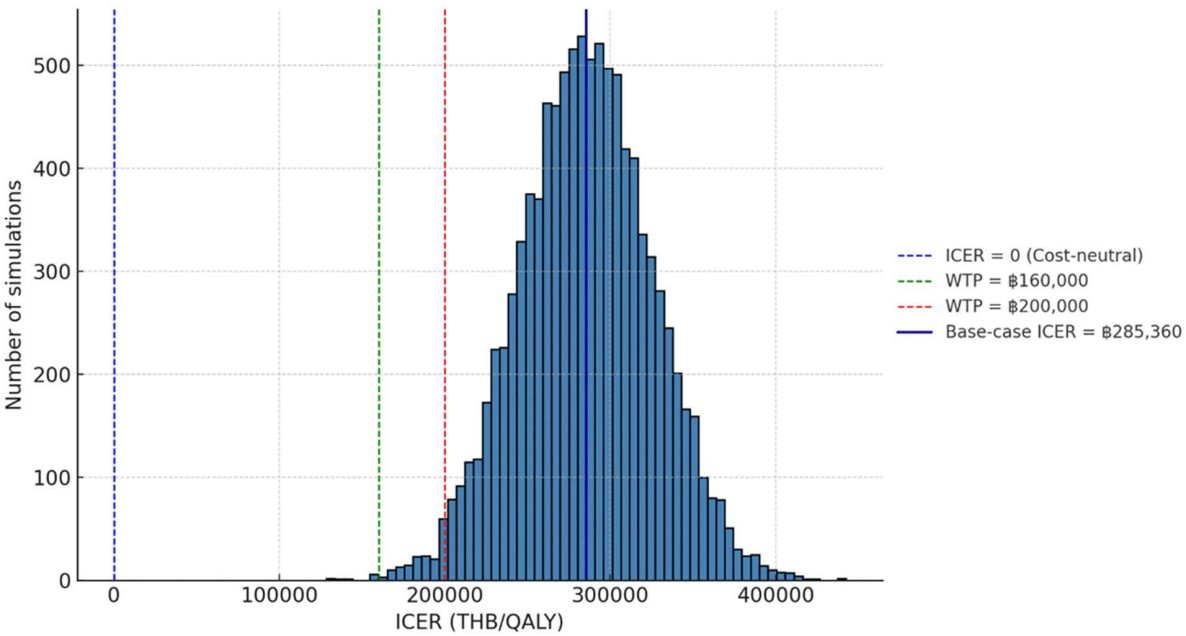

**Fig 3. Distribution of Incremental Cost-Effectiveness Ratios (ICERs) from probabilistic sensitivity analysis (PSA, per cohort of 110 patients).** The histogram displays results from 10,000 Monte Carlo simulations varying prevalence, test accuracy, cost, and utility parameters. The base-case ICER of ฿285,360/QALY is indicated (solid dark blue line), along with reference thresholds of cost-neutrality (0), Thai willingness-to-pay (WTP) at ฿160,000/QALY (green), and ฿200,000/QALY (red). Nearly all simulations exceeded the lower WTP threshold, indicating that universal screening is unlikely to be cost-effective under current cost structures. Plotted values are based on cohort totals for 110 patients; see Methods for clarification on ICER scaling. Abbreviations: ICER, incremental cost-effectiveness ratio; PSA, probabilistic sensitivity analysis; QALY, quality-adjusted life year; WTP, willingness-to-pay; THB, Thai Baht; ACE, acquired comitant esotropia; AChR-Ab, acetylcholine receptor antibody; TFT, thyroid function test; MG, myasthenia gravis; OMG, ocular myasthenia gravis.

comitant esotropia (ACE) highlights this challenge: although often regarded as idiopathic, it may be the first manifestation of systemic disease such as myasthenia gravis (MG) or thyroid eye disease (TED) [2–5,12–14]. Because atypical cases frequently lack hallmark features such as ptosis, proptosis, or lid retraction, baseline screening with acetylcholine receptor antibody (AChR-Ab) assays and thyroid function tests (TFT) has been proposed as a means of earlier detection. Yet the economic justification for universal testing has remained uncertain in Thailand, where laboratory and confirmatory costs may influence policy adoption.

This study addressed that gap by evaluating the cost-utility of universal AChR-Ab and TFT screening compared with no routine screening (symptom-triggered testing) in ACE, using a decision tree and a 10-year Markov model consistent with Thai Health Technology Assessment (HTA) standards [15,16]. When real-world diagnostic accuracy was applied, the base-case analysis produced an incremental cost-effectiveness ratio (ICER) of approximately ฿285,000 per QALY, exceeding Thailand's willingness-to-pay threshold of ฿160,000–200,000/QALY. A scenario analysis using unadjusted utilities produced more favorable cost-effectiveness estimates, reflecting optimistic assumptions rather than real-world test performance. One-way sensitivity analysis identified MG prevalence, the utility decrement from undiagnosed MG, and AChR-Ab test cost as the most influential parameters. In contrast, probabilistic sensitivity analysis confirmed that universal screening is unlikely to be cost-effective at current test prices. These findings suggest that universal AChR-Ab screening is unlikely to be economically favorable under current Thai cost structures despite potential clinical benefits. Key assumptions included a 2.7% MG prevalence, 75% sensitivity, and 98% specificity for AChR-Ab testing, as well as utility decrements derived from published literature [6,7,17]. Full parameter ranges and distributions are presented in S4 Table. While

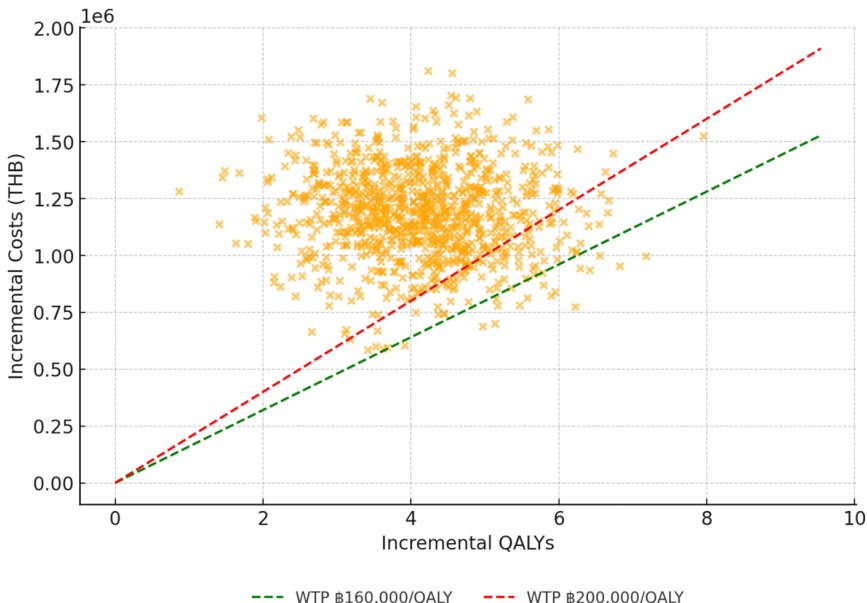

**Fig 4. Cost-Effectiveness Plane of Probabilistic Sensitivity Analysis (10,000 simulations, per cohort of 110 patients).** Each point represents one of 10,000 Monte Carlo simulations of incremental costs and QALYs for universal AChR-Ab screening compared with no routine screening (symptom-triggered testing). Dashed lines indicate Thailand's willingness-to-pay (WTP) thresholds of ฿160,000 and ฿200,000 per QALY gained. Most simulations fall above these thresholds, indicating that universal screening is not cost-effective under current cost structures. Plotted values are based on cohort totals for 110 patients; see Methods for clarification on ICER scaling. Abbreviations: AChR-Ab, acetylcholine receptor antibody; QALY, quality-adjusted life year; WTP, willingness-to-pay; THB, Thai Baht.

universal screening was not cost-effective under current cost structures, targeted screening of 30% high-risk ACE patients yielded a lower ICER (฿266,160/QALY), closer to the Thai WTP threshold.

The targeted screening strategy should be interpreted with caution. In the present model, this approach was implemented as an exploratory proportional strategy. We did not model differential disease prevalence within the targeted subgroup, nor did we incorporate a targeting mechanism with defined sensitivity or specificity. As a result, the observed improvement in cost-effectiveness reflects reduced testing volume rather than true enrichment of disease probability. Future studies should develop and validate clinical prediction rules to enable more accurate and efficient targeting of high-risk patients. Therefore, this approach should not be interpreted as a true risk-enriched diagnostic strategy, but rather as a cost-reduction scenario without validated targeting performance.

The relatively high absolute QALY values observed in the scenario analysis should be interpreted in the context of the cohort composition. Because only a small proportion of patients had MG or thyroid dysfunction, the majority of the cohort remained in health states close to full utility over the 10-year horizon. Therefore, these QALY estimates reflect population-average outcomes rather than disease-specific burden.

Our model was anchored in a cohort-derived MG prevalence of 2.7% and test performance values consistent with a large series of ocular MG [6,7]. Sensitivity estimates of 70–80% with specificity approximately 98% are higher than earlier reports of 44–66% sensitivity [7], reflecting improvements in assay technology and laboratory methods. Utilities were derived from published literature, with early-diagnosed ocular MG assigned 0.872 and delayed diagnosis 0.739 [17], capturing the quality-of-life loss from late treatment. High specificity limited false positives to a small proportion, resulting in minor costs but no sustained disutility, whereas false negatives drove the largest QALY reductions. Costs and utilities were discounted at 3% annually in line with Thai HTA guidance [15,16]. Scenario analyses with higher MG prevalence (5–10%) improved cost-effectiveness, supporting selective screening in referral centers with heavier case burdens.

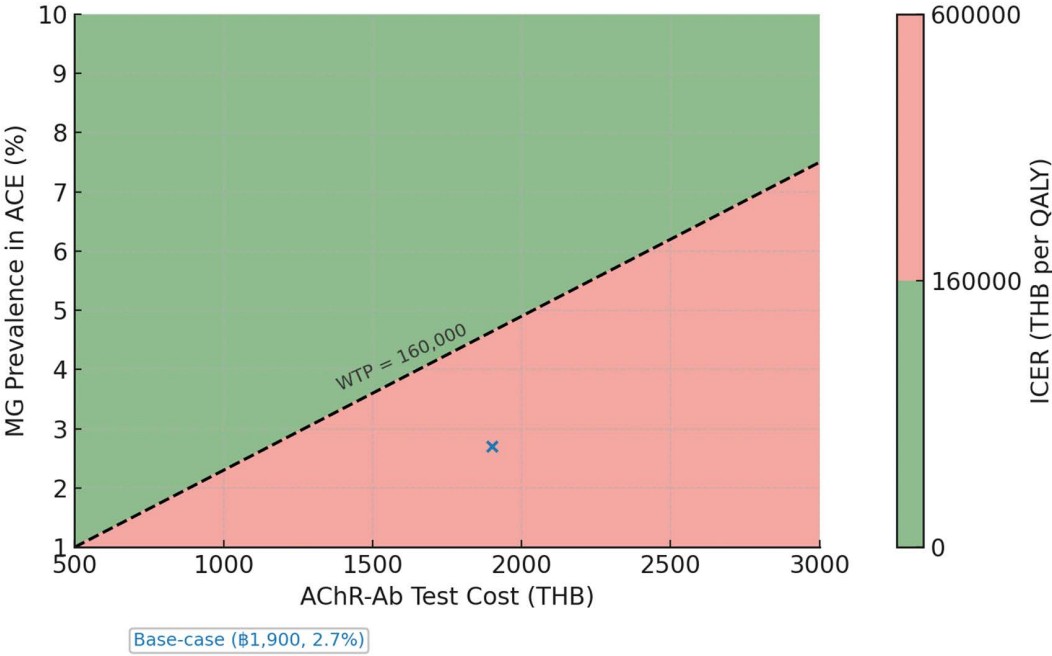

Base-case (฿1,900, 2.7%)

**Fig 5. Two-way threshold analysis of AChR-Ab test cost and MG prevalence.** The contour plot shows the incremental cost-effectiveness ratio (ICER) of universal AChR-Ab screening across varying test costs (x-axis) and MG prevalence in ACE (y-axis). Green shading indicates regions where screening is cost-effective below Thailand's willingness-to-pay (WTP) threshold of ฿160,000/QALY, while red shading indicates ICERs exceeding the threshold. The dashed line represents the WTP boundary, and the blue dot marks the base-case scenario (2.7% prevalence, ฿1900/test). Plotted values are based on cohort totals for 110 patients; see Methods for clarification on ICER scaling. Abbreviations: ACE, acquired comitant esotropia; AChR-Ab, acetylcholine receptor antibody; MG, myasthenia gravis; ICER, incremental cost-effectiveness ratio; QALY, quality-adjusted life year; WTP, willingness-to-pay; THB, Thai Baht.

ACE remains a distinct strabismus subtype characterized by sudden-onset, comitant esodeviation and diplopia, which may represent the first manifestation of systemic disease such as MG or TED even in the absence of hallmark signs [3–5,12–14]. Neuroimaging remains indispensable when neurological signs are present, with abnormal MRI reported in up to 10% of ACE cases [13,20–22]. In our institution, MRI is routinely performed to exclude structural causes, consistent with national recommendations. Within this diagnostic framework, the incremental economic value of routine AChR-Ab and TFT testing had not previously been quantified.

Recent reports confirm AChR-Ab sensitivity of 70–80% and specificity approximately 98% [6,7]. Positive serology has been associated with older age, male sex, and greater risk of generalization [6]. Tear-based assays may raise sensitivity to 80.8% [23], but serum-based testing remains the most feasible first-line option in Thailand. In our model, even with less-than-perfect sensitivity, universal screening produced QALY gains at non-trivial MG prevalence, primarily by enabling earlier treatment and reducing the risk of prolonged diplopia, strabismus progression, and systemic generalization. By contrast, TFT screening showed minimal utility, with only one patient (0.9%) diagnosed with subclinical hypothyroidism, yielding negligible QALY improvement. The low yield likely reflects the rarity of TED presenting as isolated ACE and the limited follow-up period. Selective TFT—guided by systemic hyperthyroid features, orbital signs, or demographic risk— remains a more efficient approach and is consistent with previous recommendations [8–10].

Economic differences between strategies were driven by both the costs of baseline screening and downstream costs associated with follow-up, confirmatory investigations, and treatment of detected or missed disease. Sensitivity analyses reinforced the robustness of these findings, with key parameters such as MG prevalence, utility decrement from delayed

diagnosis, and AChR-Ab cost exerting the greatest influence on model outcomes. However, the base-case ICER above Thailand's WTP threshold underscores that universal screening is not currently cost-effective. The broader implications of these findings should be interpreted cautiously, as the current analysis is preliminary and based on limited event data: although wider implementation would require investments in laboratory infrastructure, personnel, and confirmatory testing capacity, the observed savings from avoided ICU admissions and misdiagnosis-related costs (see S1 Table) may offset some of these expenditures. The exclusion of infrastructure and implementation costs likely biases the analysis toward screening and may therefore underestimate the full economic burden of universal implementation in settings without established laboratory capacity. Even if laboratory setup costs for immunoassay platforms, quality-control programs, and technician training were amortized across several thousand tests annually, the ICER would likely remain above the Thai WTP threshold, highlighting that price reductions or higher-prevalence settings are needed for cost-effectiveness. For international readers, the transferability of these findings depends strongly on local test prices. In Thailand, AChR-Ab testing costs about ฿1,900 (~$50 USD), whereas in other middle-income countries, prices may be higher due to importation or lower testing volumes. In such contexts, selective rather than universal screening may be preferable. Conversely, the case for broader use may be stronger in higher-prevalence or higher-volume systems.

Although universal AChR-Ab screening was not cost-effective under Thai WTP thresholds, the clinical value of early detection remains important. A pragmatic tiered diagnostic pathway may therefore be more appropriate: AChR-Ab testing should be considered selectively for ACE patients with higher pre-test probability (e.g., atypical clinical features or referral-center cases), followed by confirmatory neurophysiologic testing when results are positive or borderline. TFT can be reserved for those with systemic symptoms, orbital signs, or demographic risk factors. Such a targeted approach balances diagnostic yield with economic sustainability while integrating with existing MRI-based protocols.

This study has several limitations. First, the retrospective single-center design and the small number of observed outcome events (myasthenia gravis and hypothyroidism) limit the precision and stability of prevalence estimates, which are key drivers of the cost-effectiveness model and may influence ICER results. Although probabilistic sensitivity analysis addressed parameter uncertainty, it could not fully account for structural uncertainty related to model assumptions and the limited number of observed disease events. Second, health-state utility values were derived from published studies rather than Thai populations and may not fully capture long-term disease burden, including progression, treatment-related adverse effects, and variability in symptom severity. Third, the analysis was conducted from a healthcare sector perspective and included only direct medical costs. Indirect societal costs and psychological impacts, as well as capital and implementation costs (e.g., laboratory infrastructure, assay platforms, and personnel training), were not included. This exclusion likely biased the model modestly in favor of screening and may underestimate the full economic burden of universal implementation, particularly in lower-resource settings. Finally, the targeted screening strategy was modeled as an exploratory scenario without a validated risk-stratification tool. The assumption of a fixed high-risk subgroup without differential disease prevalence or targeting accuracy may oversimplify real-world clinical decision-making.

Future studies should incorporate prospective, multicenter data to improve prevalence estimates, develop population-specific utility values, and validate clinically applicable risk-stratification approaches. Emerging diagnostic modalities, such as tear-based AChR-Ab assays [23], also warrant further evaluation in cost-effectiveness frameworks.

## Conclusion

In the Thai tertiary-care context, universal AChR-Ab screening in ACE may confer clinical benefit through earlier detection of MG, but it was not cost-effective in the present base-case analysis. Selective screening in higher-risk presentations and symptom-triggered thyroid testing may represent promising strategies for further evaluation within existing healthcare infrastructures but require confirmation in larger studies before consideration for policy implementation.

## Supporting information

**S1 Fig. Decision Tree Comparing Diagnostic Strategies in Acquired Comitant Esotropia (ACE).** This decision tree illustrates two diagnostic strategies for patients with acquired comitant esotropia (ACE): (1) no routine laboratory screening (symptom-triggered testing), and (2) universal baseline screening with acetylcholine receptor antibody (AChR-Ab) and thyroid function tests (TFT; TSH, FT3, FT4). Branch probabilities were derived from cohort prevalence (ocular myasthenia gravis [OMG], 2.7%; thyroid dysfunction, 0.9%), published diagnostic performance data (AChR-Ab sensitivity 0.75, specificity 0.98; TFT sensitivity 0.90, specificity 0.92), and observed outcomes. Terminal nodes indicate health state utilities and corresponding costs in 2024 Thai Baht (THB), discounted at 3% per year. True positives received early treatment and achieved higher quality-adjusted life years (QALYs); false negatives experienced delayed or missed diagnosis with lower utility; false positives incurred additional confirmatory investigations and follow-up costs; and true negatives were managed as idiopathic ACE. Abbreviations: ACE, acquired comitant esotropia; AChR-Ab, acetylcholine receptor antibody; TFT, thyroid function test; TSH, thyroid-stimulating hormone; FT3, free triiodothyronine; FT4, free thyroxine; MG, myasthenia gravis; OMG, ocular myasthenia gravis; TED, thyroid eye disease; TP, true positive; TN, true negative; FP, false positive; FN, false negative; QALY, quality-adjusted life year; THB, Thai Baht.
(TIF)

**S2 Fig. Simplified Markov Model for ACE Screening.** The Markov model represents 10-year transitions among long-term health states in ACE. Initial states include idiopathic ACE, ocular myasthenia gravis (OMG), and hypothyroid or thyroid eye disease (TED). Patients with OMG may remain stable, progress to generalized MG, or die. Those with hypothyroidism/TED may remain in the same state or die. Idiopathic ACE cases may remain stable, newly develop OMG or hypothyroidism, or die due to background age-related mortality. Mortality was modeled as an absorbing state representing all-cause death rather than ACE-specific fatality. Abbreviations: ACE, acquired comitant esotropia; OMG, ocular myasthenia gravis; MG, myasthenia gravis; TED, thyroid eye disease; THY, hypothyroid/TED combined state.
(TIF)

**S3 Fig. Cost-Effectiveness Acceptability Curve (CEAC) for Universal AChR-Ab Screening (Per Cohort of 110 Patients).** The CEAC displays the probability that universal acetylcholine receptor antibody (AChR-Ab) screening is cost-effective compared with no routine screening (symptom-triggered testing) across a range of willingness-to-pay (WTP) thresholds, based on 10,000 Monte Carlo simulations from probabilistic sensitivity analysis (PSA). Vertical dashed lines mark Thailand's reference WTP thresholds of ฿160,000 and ฿200,000 per quality-adjusted life year (QALY). At these levels, the probability of cost-effectiveness remains low, reflecting the base-case incremental cost-effectiveness ratio (ICER) of approximately ฿285,360/QALY. The probability increases only at higher thresholds, indicating that universal screening is unlikely to be cost-effective under current Thai cost structures. Abbreviations: ACE, acquired comitant esotropia; AChR-Ab, acetylcholine receptor antibody; CEAC, cost-effectiveness acceptability curve; ICER, incremental cost-effectiveness ratio; PSA, probabilistic sensitivity analysis; QALY, quality-adjusted life year; THB, Thai Baht; WTP, willingness-to-pay.
(TIF)

**S1 Table. Clinical and Diagnostic Characteristics of Patients with Positive Laboratory Findings.** This table details presenting features, laboratory and imaging findings, final diagnoses, and early treatment outcomes for patients with abnormal AChR-Ab or TFT results. Abbreviations: CT, computed tomography; MRI, magnetic resonance imaging; MG, myasthenia gravis; OMG, ocular myasthenia gravis; F, female; TSH, thyroid-stimulating hormone; FT3, free triiodothyronine; FT4, free thyroxine.
(DOCX)

**S2 Table. Utility Weights for Myasthenia Gravis (MG), Ocular Myasthenia Gravis (OMG), and Thyroid Disorders.** This table summarizes the health utility values applied to estimate quality-adjusted life years (QALYs) across diagnostic and treatment states in the economic model. Abbreviations: MG, myasthenia gravis; OMG, ocular myasthenia gravis; ADL, activities of daily living; TED, thyroid eye disease; QALY, quality-adjusted life year.
(DOCX)

**S3 Table. Cost Components by Diagnostic Strategy (10-Year Horizon, 3% Discount).** Costs are presented in 2024 Thai Baht (THB), discounted at 3% annually following Thai Health Technology Assessment (HTA) guidelines. Unit costs were obtained from national hospital billing data and adjusted for inflation using the Thai consumer price index (CPI). Quantities reflect observed case prevalence and modeled cohort size (n = 110). Abbreviations: AChR-Ab, acetylcholine receptor antibody; TFT, thyroid function test; TSH, thyroid-stimulating hormone; FT3, free triiodothyronine; FT4, free thyroxine; MRI, magnetic resonance imaging; CT, computed tomography; THB, Thai Baht; TP, true positive; FP, false positive; OMG, ocular myasthenia gravis.
(DOCX)

**S4 Table. Model Input Parameters for Cost–Utility Analysis.** This table summarizes the input assumptions for epidemiologic, diagnostic, cost, and utility variables used in the decision-analytic model. Parameter ranges reflect ±20% variation from base-case estimates unless otherwise specified. For probabilistic sensitivity analysis (PSA), beta distributions were assigned to probabilities and utilities, and gamma distributions to cost parameters. Abbreviations: ACE, acquired comitant esotropia; AChR-Ab, acetylcholine receptor antibody; HTA, health technology assessment; MG, myasthenia gravis; OMG, ocular myasthenia gravis; QALY, quality-adjusted life year; TFT, thyroid function test; TSH, thyroid-stimulating hormone; FT3, free triiodothyronine; FT4, free thyroxine; THB, Thai Baht; WTP, willingness-to-pay.
(DOCX)

**S4a Table. Distribution Parameters for Probabilistic Sensitivity Analysis.** Probabilities and utilities were modeled using beta distributions, parameterized by $\alpha = p \times n$ and $\beta = (1 - p) \times n$. For instance, a sensitivity of 75% (p = 0.75) with an assumed sample size of n = 100 yields $\alpha = 75$, $\beta = 25$. Cost parameters followed gamma distributions, defined by shape ($k = \mu^2/\sigma^2$) and scale ($\theta = \sigma^2/\mu$), assuming variance ±20% of the mean in the absence of empirical data. This method adheres to standard practice for health economic modeling and PSA in line with Thai HTA recommendations.
(DOCX)

**S1 File. CHEERS 2022 Checklist.** Completed Consolidated Health Economic Evaluation Reporting Standards (CHEERS 2022) checklist for this study, indicating where each reporting item is addressed in the manuscript.
(DOCX)

**S2 File. De-identified study dataset.** An anonymized patient-level dataset used for the cost–utility analysis, including demographic information, diagnostic investigations, treatment variables, and model input data. All direct identifiers were removed prior to sharing to protect participant confidentiality.
(XLSX)

## Acknowledgments

I express my sincere gratitude to Dr. Thitiporn Ratanapojnart, Chief Director of the Ophthalmology Department at Phramongkutklao Hospital, for her invaluable support and guidance throughout this study. Her expertise and leadership were instrumental in the successful completion of this research.

## Author contributions

**Conceptualization:** Worapot Srimanan.

**Data curation:** Worapot Srimanan, Phawasutthi Keokajee.

**Formal analysis:** Worapot Srimanan, Phawasutthi Keokajee, Sunita Sawangsribanterng.

**Funding acquisition:** Worapot Srimanan.

**Investigation:** Worapot Srimanan, Sunita Sawangsribanterng.

**Methodology:** Worapot Srimanan, Sunita Sawangsribanterng.

**Project administration:** Worapot Srimanan.

**Resources:** Worapot Srimanan.

**Software:** Worapot Srimanan.

**Supervision:** Worapot Srimanan.

**Validation:** Worapot Srimanan.

**Visualization:** Worapot Srimanan.

**Writing – original draft:** Worapot Srimanan, Phawasutthi Keokajee, Sunita Sawangsribanterng.

**Writing – review & editing:** Worapot Srimanan, Sunita Sawangsribanterng.

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
