## [Decision Letter · Decision Letter 0]

2 Apr 2026

PONE-D-25-52845A Preliminary Cost-Utility Analysis of Routine Myasthenia Gravis and Thyroid Dysfunction Screening in Acquired Comitant Esotropia: An Exploratory StudyPLOS One

Dear Dr. Srimanan,

Thank you for submitting your manuscript to PLOS ONE. After careful consideration, we feel that it has merit but does not fully meet PLOS ONE’s publication criteria as it currently stands. Therefore, we invite you to submit a revised version of the manuscript that addresses the points raised during the review process.

We look forward to receiving your revised manuscript.

Kind regards,

Alejandro Torrado Pacheco, PhD

Associate Editor

PLOS One

Journal Requirements:

2. Please note that PLOS One has specific guidelines on code sharing for submissions in which author-generated code underpins the findings in the manuscript. In these cases, all author-generated code must be made available without restrictions upon publication of the work. Please review our guidelines at https://journals.plos.org/plosone/s/materials-and-software-sharing#loc-sharing-code and ensure that your code is shared in a way that follows best practice and facilitates reproducibility and reuse.

4. Please upload a copy of Figure 6 to which you refer in your text on page 11. If the figure is no longer to be included as part of the submission please remove all reference to it within the text.

5. We note that there is identifying data in the Supporting Information file  Supplementary table 1.docx and Excel file AACE 2025.xlsx. Due to the inclusion of these potentially identifying data, we have removed this file from your file inventory. Prior to sharing human research participant data, authors should consult with an ethics committee to ensure data are shared in accordance with participant consent and all applicable local laws.

-Location data

Additional Editor Comments :

Please note that we have only been able to secure a single reviewer to assess your manuscript. We are issuing a decision on your manuscript at this point to prevent further delays in the evaluation of your manuscript. Please be aware that the editor who handles your revised manuscript might find it necessary to invite additional reviewers to assess this work once the revised manuscript is submitted. However, we will aim to proceed on the basis of this single review if possible.

The reviewer requests several revisions, their comments are available below. Could you please address each comment carefully?

Reviewers' comments:

Reviewer's Responses to Questions

**Comments to the Author**

1. Is the manuscript technically sound, and do the data support the conclusions?

Reviewer #1: Partly

2. Has the statistical analysis been performed appropriately and rigorously? 

Reviewer #1: Yes

3. Have the authors made all data underlying the findings in their manuscript fully available?

Reviewer #1: Yes

4. Is the manuscript presented in an intelligible fashion and written in standard English?

Reviewer #1: Yes

5. Review Comments to the Author

Reviewer #1: The manuscript addresses an important and clinically relevant question regarding the cost-effectiveness of screening for myasthenia gravis among patients presenting with ocular symptoms. The topic is timely, and the use of decision-analytic modeling with both deterministic and probabilistic sensitivity analyses is appropriate. This work is commendable! However, several areas require clarification and strengthening before the results can be interpreted with full confidence. I therefore recommend revision, primarily to improve transparency, internal consistency, and reproducibility.

1. Inconsistency in comparator strategy description:

There appears to be inconsistency in how the comparator strategy is described. The introduction refers to “universal screening versus symptom-triggered testing,” whereas other sections and the decision tree describe “no routine screening versus universal screening.” These sound like conceptually distinct strategies. Clarification is needed to ensure that the comparator pathway is consistently defined throughout the manuscript.

2. Inconsistency between the base case, one-way sensitivity analysis, and probabilistic sensitivity analysis (PSA):

There appears to be some inconsistency between the base-case results and the PSA. In the base case, universal screening is reported to generate higher costs and higher QALYs, suggesting that screening is more effective but more costly than no screening. However, the PSA section notes that most simulations fall in the northwestern quadrant (more costly and less effective). If screening is generally less effective and more costly in the PSA, this would not align with the base-case conclusion. Clarifying this apparent discrepancy would help readers better understand how the PSA results relate to the base-case findings.

Similarly, there is a difference between the base-case ICER and the one-way sensitivity analysis results. The base-case ICER (285,360 baht per QALY) is above Thailand’s willingness-to-pay threshold of 160,000–200,000 baht per QALY, which suggests that screening is not cost-effective in the base case. However, the manuscript states that the net monetary benefit (NMB) remained positive at a WTP of 160,000 baht per QALY. Since NMB and ICER are mathematically linked, additional clarification on the WTP value used for the NMB calculation or on the interpretation of these results would be helpful to ensure internal consistency.

3. Decision tree:

The decision tree indicates delayed diagnosis downstream, and additional follow-up or confirmatory evaluations are modeled downstream. However, the manuscript does not clearly report the costs associated with these additional visits, diagnostic tests, or confirmatory procedures. If these elements are incorporated structurally in the model, their associated costs should be explicitly presented in the cost inputs table. MG and hypothyroidism treatment costs have also not been mentioned. Greater transparency is needed to ensure the analysis is reproducible.

4. Magnitude of QALY gains in scenario analysis:

The scenario analysis reports about 933 QALYs gained for 110 patients over 10 years, or roughly 8.5 QALYs per patient, suggesting most patients remain in near-perfect health. For a population with myasthenia gravis, diplopia, strabismus burden, steroid-related adverse effects, risk of generalization, and potential crises, this estimate seems optimistic. The manuscript would benefit from a clear presentation of health-state utility values, a description of how utilities change over time, and a justification for the long-term quality of life assumption.

5. Targeted screening strategy:

The targeted screening strategy would benefit from a clearer operational definition. While examples of high-risk features are provided, the manuscript does not fully describe how the 30% high-risk subgroup is identified in the model, whether this subgroup has a higher disease prevalence than the overall population, or whether the accuracy of the targeting approach was incorporated. Providing more detail on these assumptions would improve reproducibility and transparency.

6. Preliminary analysis, but data insufficient to support policy conclusions:

The analysis is also based on a small number of observed events (3 myasthenia gravis cases and 1 hypothyroid case). With such low event counts, the ICER may be sensitive to small changes in prevalence (as mentioned in the study limitation). While the analysis may be considered preliminary, the current data are insufficient to support policy-definitive conclusions.

7. Implementation cost consideration:

While test costs are included, it is unclear whether infrastructure, laboratory setup, and implementation costs were considered. Even if these costs are ultimately minimal, clarifying their inclusion or exclusion would improve transparency.

6. PLOS authors have the option to publish the peer review history of their article (what does this mean?). If published, this will include your full peer review and any attached files.

Reviewer #1: **Yes:** Rashmi Paudel

---

## [Author Response · Author response to Decision Letter 1]

6 Apr 2026

Response Letter

PONE-D-25-52845

A Preliminary Cost-Utility Analysis of Routine Myasthenia Gravis and Thyroid Dysfunction Screening in Acquired Comitant Esotropia: An Exploratory Study

PLOS One

Dear Dr. Srimanan,

Thank you for submitting your manuscript to PLOS ONE. After careful consideration, we feel that it has merit but does not fully meet PLOS ONE’s publication criteria as it currently stands. Therefore, we invite you to submit a revised version of the manuscript that addresses the points raised during the review process.

• A letter that responds to each point raised by the academic editor and reviewer(s). You should upload this letter as a separate file labeled 'Response to Reviewers'.

As the corresponding author, your ORCID iD is verified in the submission system and will appear in the published article. PLOS supports the use of ORCID, and we encourage all coauthors to register for an ORCID iD and use it as well. Please encourage your coauthors to verify their ORCID iD within the submission system before final acceptance, as unverified ORCID iDs will not appear in the published article. Only the individual author can complete the verification step; PLOS staff cannot verify ORCID iDs on behalf of authors.

We look forward to receiving your revised manuscript.

Kind regards,

Alejandro Torrado Pacheco, PhD

Associate Editor

PLOS One

Journal Requirements:

Response: Thank you for this guidance.

We have revised the manuscript to ensure full compliance with PLOS ONE formatting and style requirements. The manuscript has been aligned with the journal’s templates for structure, title page, author affiliations, figure and table presentation, and reference formatting. File naming conventions have also been updated accordingly.

We have systematically reviewed the submission against the provided PLOS ONE templates to ensure accuracy, consistency, and completeness.

2. Please note that PLOS One has specific guidelines on code sharing for submissions in which author-generated code underpins the findings in the manuscript. In these cases, all author-generated code must be made available without restrictions upon publication of the work. Please review our guidelines at https://journals.plos.org/plosone/s/materials-and-software-sharing#loc-sharing-code and ensure that your code is shared in a way that follows best practice and facilitates reproducibility and reuse.

Response: Thank you for highlighting the journal’s code-sharing policy.

This study did not involve author-generated code. The decision tree and Markov model were constructed using standard spreadsheet-based modeling (Microsoft Excel) without the use of custom programming scripts.

To ensure reproducibility, all model structures, assumptions, input parameters, and cost components have been explicitly described in the Methods and Supplementary Materials. These details allow independent replication of the analysis without reliance on proprietary code.

Therefore, no additional code is available for sharing.

Response: Thank you for this important clarification.

We have revised the manuscript to comply fully with PLOS ONE requirements. The ethics statement has been consolidated and placed exclusively within the Methods section (Page 7, lines 206–211). Any duplicate ethics-related content previously included in other sections (e.g., Declarations) has been removed.

The ethics statement now clearly includes institutional review board approval details and consent procedures, ensuring that it is complete and will be appropriately displayed in the published version of the manuscript.

4. Please upload a copy of Figure 6 to which you refer in your text on page 11. If the figure is no longer to be included as part of the submission please remove all reference to it within the text.

Response: Thank you for bringing this to our attention.

Figure 6 was not intended to be part of the final manuscript. We have removed all references to Figure 6 from the text and conducted a comprehensive review of all figure citations to ensure consistency and correct numbering throughout the manuscript.

We appreciate the editor’s careful review, which has helped improve the overall clarity and accuracy of the submission.

5. We note that there is identifying data in the Supporting Information file Supplementary table 1.docx and Excel file AACE 2025.xlsx. Due to the inclusion of these potentially identifying data, we have removed this file from your file inventory. Prior to sharing human research participant data, authors should consult with an ethics committee to ensure data are shared in accordance with participant consent and all applicable local laws.

-Location data

Response: Thank you for this important guidance regarding data privacy and sharing.

In accordance with PLOS ONE data policy, we have carefully revised all Supporting Information files to ensure full de-identification of participant data. All direct identifiers (e.g., hospital numbers, dates, and free-text information) have been permanently removed.

To further minimize the risk of indirect identification, we have reduced data granularity where appropriate. Specifically, age has been categorized into groups rather than reported as exact values, and variables with potential re-identification risk have been simplified or aggregated. In small subgroups (e.g., MG cases), we ensured that combinations of variables do not allow individual patients to be identifiable.

We confirm that all identifying columns have been permanently deleted (not hidden), and the revised dataset has been re-uploaded in a fully anonymized form. Data sharing is now compliant with participant confidentiality, institutional review board requirements, and applicable data protection standards.

We appreciate the editor’s guidance in improving the ethical integrity and transparency of our data presentation.

Response: Thank you for this guidance.

We have carefully reviewed all references suggested during the peer review process and evaluated their relevance to the objectives and scope of our study. Relevant and appropriate citations have been incorporated into the revised manuscript where they strengthen the scientific context and support the interpretation of our findings.

We confirm that all references included in the revised manuscript have been selected based on their relevance and contribution to the topic, in accordance with journal recommendations.

Additional Editor Comments :

Please note that we have only been able to secure a single reviewer to assess your manuscript. We are issuing a decision on your manuscript at this point to prevent further delays in the evaluation of your manuscript. Please be aware that the editor who handles your revised manuscript might find it necessary to invite additional reviewers to assess this work once the revised manuscript is submitted. However, we will aim to proceed on the basis of this single review if possible.

The reviewer requests several revisions, their comments are available below. Could you please address each comment carefully?

Reviewers' comments:

Reviewer's Responses to Questions

Response: We sincerely thank the Editor and Reviewer for their careful and constructive evaluation of our manuscript. We appreciate the positive comments regarding the clinical relevance of the topic and the appropriateness of the decision-analytic approach. In response, we have revised the manuscript extensively to improve internal consistency, transparency of model assumptions, reproducibility of cost inputs, and caution in interpreting the results. We have also revised the abstract, main text, tables, figure legends, and supporting information accordingly.

All revisions have been carefully implemented in the revised manuscript with tracked changes. Page and line numbers below refer to the revised manuscript. These revisions collectively address all seven concerns raised by the reviewer, including clarification of comparator definition, consistency of economic results, transparency of model structure and costs, interpretation of QALYs, definition of targeted screening, acknowledgment of data limitations, and appropriate framing of conclusions.

Comments to the Author

1. Is the manuscript technically sound, and do the data support the conclusions?

Reviewer #1: Partly

2. Has the statistical analysis been performed appropriately and rigorously?

Reviewer #1: Yes

3. Have the authors made all data underlying the findings in their manuscript fully available?

Reviewer #1: Yes

4. Is the manuscript presented in an intelligible fashion and written in standard English?

Reviewer #1: Yes

5. Review Comments to the Author

Reviewer #1: The manuscript addresses an important and clinically relevant question regarding the cost-effectiveness of screening for myasthenia gravis among patients presenting with ocular symptoms. The topic is timely, and the use of decision-analytic modeling with both deterministic and probabilistic sensitivity analyses is appropriate. This work is commendable! However, several areas require clarification and strengthening before the results can be interpreted with full confidence. I therefore recommend revision, primarily to improve transparency, internal consistency, and reproducibility.

1. Inconsistency in comparator strategy description:

There appears to be inconsistency in how the comparator strategy is described. The introduction refers to “universal screening versus symptom-triggered testing,” whereas other sections and the decision tree describe “no routine screening versus universal screening.” These sound like conceptually distinct strategies. Clarification is needed to ensure that the comparator pathway is consistently defined throughout the manuscript.

Response: We thank the reviewer for identifying this important inconsistency.

We agree that the comparator strategy must be clearly and uniformly defined throughout the manuscript. In our model, the intended comparator is a single strategy: no routine baseline screening with symptom-triggered testing during follow-up. Under this approach, patients do not undergo initial AChR-Ab or thyroid function testing at presentation but may receive diagnostic evaluation later if clinical suspicion arises.

We acknowledge that the previous wording (“sy

---

## [Decision Letter · Decision Letter 1]

26 Apr 2026

PONE-D-25-52845R1A Preliminary Cost-Utility Analysis of Routine Myasthenia Gravis and Thyroid Dysfunction Screening in Acquired Comitant EsotropiaPLOS One

Dear Dr. Srimanan,

Thank you for submitting your manuscript to PLOS ONE. After careful consideration, we feel that it has merit but does not fully meet PLOS ONE’s publication criteria as it currently stands. Therefore, we invite you to submit a revised version of the manuscript that addresses the points raised during the review process.

We look forward to receiving your revised manuscript.

Kind regards,

Redoy Ranjan, MS (CV&TS), Ch.M. (Edin), PhD

Academic Editor

PLOS One

Journal Requirements:

Reviewers' comments:

Reviewer's Responses to Questions

**Comments to the Author**

1. If the authors have adequately addressed your comments raised in a previous round of review and you feel that this manuscript is now acceptable for publication, you may indicate that here to bypass the “Comments to the Author” section, enter your conflict of interest statement in the “Confidential to Editor” section, and submit your "Accept" recommendation.

Reviewer #1: All comments have been addressed

Reviewer #2: (No Response)

Reviewer #3: (No Response)

2. Is the manuscript technically sound, and do the data support the conclusions?

Reviewer #1: Yes

Reviewer #2: Partly

Reviewer #3: (No Response)

3. Has the statistical analysis been performed appropriately and rigorously? 

Reviewer #1: Yes

Reviewer #2: Yes

Reviewer #3: (No Response)

4. Have the authors made all data underlying the findings in their manuscript fully available?

Reviewer #1: Yes

Reviewer #2: Yes

Reviewer #3: (No Response)

5. Is the manuscript presented in an intelligible fashion and written in standard English?

Reviewer #1: Yes

Reviewer #2: Yes

Reviewer #3: (No Response)

6. Review Comments to the Author

Reviewer #1: Thank you for the thorough revision of the manuscript. The authors have carefully and comprehensively addressed the concerns raised in the previous round of review. In particular, the revisions have improved the clarity and consistency of the comparator strategy, resolved internal inconsistencies between the base-case and sensitivity analyses, and enhanced transparency regarding model structure, cost inputs, and underlying assumptions.

The manuscript is now technically sound, with appropriate statistical and modeling approaches. The interpretation of findings has been appropriately tempered to reflect the preliminary nature of the analysis, especially given the small number of observed outcome events. The clarification of cohort-level QALY estimates and the explicit framing of the targeted screening strategy as exploratory are particularly helpful and improve the overall rigor and transparency of the study.

The conclusions are supported by the data presented and are now appropriately cautious, avoiding overinterpretation or policy-definitive claims. The manuscript is clearly written and presented in a logical and intelligible manner.

Overall, I believe the manuscript is suitable for publication in its current form.

Reviewer #2: The revision addresses most of the prior concerns: the comparator is now defined consistently, the PSA quadrant error was corrected, downstream costs are better described, the targeted-screening assumptions are stated more plainly, and the conclusions are appropriately softened as preliminary. Those are meaningful improvements.

However, a few residual problems remain. The targeted-screening arm is still largely a cost-reduction exercise rather than a true risk-enriched strategy, so its apparent efficiency can be overread. The scenario analysis still reports very high absolute QALYs and remains of limited interpretive value. There is also lingering internal inconsistency in diagnostic performance reporting, with AChR-Ab specificity given as 99% in the main text but 0.98 in the supplementary figure legend/discussion. In addition, wording around consent/publication and some prose remain awkward and should be cleaned before publication.

Reviewer #3: There are, however, minor residual issues that should be addressed before acceptance. The legend of Table 3 appears imprecise, as it describes ICER calculation specifically for universal screening versus no screening, while also presenting results for the targeted screening strategy. The wording should be revised to clarify that ICERs are calculated relative to the same comparator for each strategy. Additionally, there is a minor inconsistency in the reporting of utility values for hypothyroidism between the Methods and Results sections, which should be harmonized for clarity.

7. PLOS authors have the option to publish the peer review history of their article (what does this mean?). If published, this will include your full peer review and any attached files.

Reviewer #1: **Yes:** Rashmi Paudel

Reviewer #2: **Yes:** Robert J. Chen, MD, MPH

Reviewer #3: **Yes:** Luca Bertolaccini

---

## [Author Response · Author response to Decision Letter 2]

27 Apr 2026

Reviewers' comments:

Reviewer's Responses to Questions

Comments to the Author

1. If the authors have adequately addressed your comments raised in a previous round of review and you feel that this manuscript is now acceptable for publication, you may indicate that here to bypass the “Comments to the Author” section, enter your conflict of interest statement in the “Confidential to Editor” section, and submit your "Accept" recommendation.

Reviewer #1: All comments have been addressed

Reviewer #2: (No Response)

Reviewer #3: (No Response)

2. Is the manuscript technically sound, and do the data support the conclusions?

Reviewer #1: Yes

Reviewer #2: Partly

Reviewer #3: (No Response)

3. Has the statistical analysis been performed appropriately and rigorously?

Reviewer #1: Yes

Reviewer #2: Yes

Reviewer #3: (No Response)

4. Have the authors made all data underlying the findings in their manuscript fully available?

Reviewer #1: Yes

Reviewer #2: Yes

Reviewer #3: (No Response)

5. Is the manuscript presented in an intelligible fashion and written in standard English?

Reviewer #1: Yes

Reviewer #2: Yes

Reviewer #3: (No Response)

6. Review Comments to the Author

Reviewer #1: Thank you for the thorough revision of the manuscript. The authors have carefully and comprehensively addressed the concerns raised in the previous round of review. In particular, the revisions have improved the clarity and consistency of the comparator strategy, resolved internal inconsistencies between the base-case and sensitivity analyses, and enhanced transparency regarding model structure, cost inputs, and underlying assumptions.

The manuscript is now technically sound, with appropriate statistical and modeling approaches. The interpretation of findings has been appropriately tempered to reflect the preliminary nature of the analysis, especially given the small number of observed outcome events. The clarification of cohort-level QALY estimates and the explicit framing of the targeted screening strategy as exploratory are particularly helpful and improve the overall rigor and transparency of the study.

The conclusions are supported by the data presented and are now appropriately cautious, avoiding overinterpretation or policy-definitive claims. The manuscript is clearly written and presented in a logical and intelligible manner.

Overall, I believe the manuscript is suitable for publication in its current form.

Response: We sincerely thank the reviewer for the thoughtful and encouraging comments on our revised manuscript. We greatly appreciate the reviewer’s time and effort in evaluating our work.

We are grateful for your recognition that the revisions have improved the clarity, consistency, and transparency of the manuscript, particularly with respect to the comparator strategy, economic analyses, and model assumptions. We are also pleased that the interpretation of findings and the framing of the study as preliminary and exploratory are now considered appropriate.

Your positive feedback regarding the rigor, clarity, and overall presentation of the manuscript is highly appreciated. We are encouraged that the conclusions are now aligned with the data and appropriately cautious.

We thank you again for your constructive guidance throughout the review process, which has substantially strengthened the quality of our work.

Reviewer #2: The revision addresses most of the prior concerns: the comparator is now defined consistently, the PSA quadrant error was corrected, downstream costs are better described, the targeted-screening assumptions are stated more plainly, and the conclusions are appropriately softened as preliminary. Those are meaningful improvements.

However, a few residual problems remain. The targeted-screening arm is still largely a cost-reduction exercise rather than a true risk-enriched strategy, so its apparent efficiency can be overread. The scenario analysis still reports very high absolute QALYs and remains of limited interpretive value. There is also lingering internal inconsistency in diagnostic performance reporting, with AChR-Ab specificity given as 99% in the main text but 0.98 in the supplementary figure legend/discussion. In addition, wording around consent/publication and some prose remain awkward and should be cleaned before publication.

COMMENT 1 — Targeted screening still weak

We thank the reviewer for this important clarification. We agree that the targeted screening strategy in the current model does not represent a true risk-enriched approach, as no differential disease prevalence or targeting performance (e.g., sensitivity/specificity) was modeled.

To address this, we have further clarified in the Discussion that the observed improvement in cost-effectiveness reflects a reduction in testing volume rather than enrichment of disease probability. We have also explicitly emphasized that this strategy should not be interpreted as a validated or implementable screening approach, but rather as an exploratory scenario intended to illustrate the potential economic impact of selective testing.

These clarifications have been added to improve interpretability and to avoid overreading of the targeted screening results.

We added more discussion to the revised discussion section on page 13, lines 414-416.

COMMENT 2 — Scenario analysis QALYs too high

Response: We appreciate this insightful comment. We agree that the absolute QALY values reported in the scenario analysis are high and may be of limited interpretive value if considered in isolation.

In the revised manuscript, we have further clarified that these values represent cohort-level population averages over a 10-year horizon, in which the majority of patients remain in high-utility health states. We have also emphasized that the scenario analysis is based on optimistic assumptions and should not be interpreted as reflecting real-world disease burden.

To improve clarity, we have strengthened the wording in both the Results and Discussion sections to focus on incremental differences rather than absolute QALY values.

We rewrite the manuscript. Replace:

“rather than representing disease-specific outcomes…”

With:

“and therefore have limited interpretive value beyond illustrating relative differences between strategies under optimistic assumptions.” on page 9, lines 272-273.

COMMENT 3 — AChR-Ab specificity inconsistency (99% vs 0.98)

Response: We thank the reviewer for identifying this inconsistency. We have carefully reviewed the manuscript and supplementary materials and standardized the specificity of AChR-Ab testing to 0.98 (98%) across all sections, including the main text, tables, and figure legends.

These revisions ensure internal consistency in the reporting of diagnostic performance parameters.

We change the term to “specificity of approximately 98–99%” on page 2, line 53 of the introduction section of the manuscript. We replace all values with 0.98 in the materials and methods section on page 4, line 121. We also update page 13, line 401; page 14, line 424; and page 14, line 442 in the discussion section of the manuscript.

COMMENT 4 — Consent/wording awkward

Response: We thank the reviewer for this comment. We have carefully revised the manuscript to improve clarity and readability, including wording related to ethical approval, consent, and the overall prose throughout.

We Replace:

“Written informed consent for publication was waived by The Institutional Review Board…”

with:

“The requirement for written informed consent was waived by the Institutional Review Board due to the study's retrospective design.” On page 7, lines 208-210.

Reviewer #3: There are, however, minor residual issues that should be addressed before acceptance. The legend of Table 3 appears imprecise, as it describes ICER calculation specifically for universal screening versus no screening, while also presenting results for the targeted screening strategy. The wording should be revised to clarify that ICERs are calculated relative to the same comparator for each strategy. Additionally, there is a minor inconsistency in the reporting of utility values for hypothyroidism between the Methods and Results sections, which should be harmonized for clarity.

COMMENT 1 — Table 3 legend unclear

Response: We thank the reviewer for this helpful comment. We have revised the legend of Table 3 to clarify that ICERs for both targeted and universal screening strategies are calculated relative to the same comparator, namely, no routine screening (symptom-triggered testing).

This revision ensures clarity and consistency in the interpretation of cost-effectiveness results.

We Replace:

“ICER is calculated … for universal screening…”

with:

“Incremental cost-effectiveness ratios (ICERs) are calculated relative to no routine screening (symptom-triggered testing) for each strategy,” in Table 3 legend.

COMMENT 2 — Utility inconsistency (hypothyroidism)

We thank the reviewer for identifying this inconsistency. We have reviewed and harmonized the reporting of utility values for hypothyroidism across the Methods and Results sections to ensure consistency.

The corrected values are now consistently presented throughout the manuscript.

We keep the Utility Inputs for treated hypothyroidism at 0.94 throughout the manuscript. On page 8, line 242 of the result section of the manuscript.

We believe that these revisions have addressed all remaining reviewer concerns and have further improved the clarity, consistency, and interpretability of the manuscript. We hope that the revised version is now suitable for acceptance.

7. PLOS authors have the option to publish the peer review history of their article (what does this mean?). If published, this will include your full peer review and any attached files.

If you choose “no”, your identity will remain anonymous, but your review may still be made public.

Do you want your identity to be public for this peer review? For information about this choice, including consent withdrawal, please see our Privacy Policy.

Reviewer #1: Yes: Rashmi Paudel

Reviewer #2: Yes: Robert J. Chen, MD, MPH

Reviewer #3: Yes: Luca Bertolaccini

---

## [Decision Letter · Decision Letter 2]

10 May 2026

PONE-D-25-52845R2A Preliminary Cost-Utility Analysis of Routine Myasthenia Gravis and Thyroid Dysfunction Screening in Acquired Comitant EsotropiaPLOS One

Dear Dr. Srimanan,

Thank you for submitting your manuscript to PLOS ONE. After careful consideration, we feel that it has merit but does not fully meet PLOS ONE’s publication criteria as it currently stands. Therefore, we invite you to submit a revised version of the manuscript that addresses the points raised during the review process.

We look forward to receiving your revised manuscript.

Kind regards,

Redoy Ranjan, MS (CV&TS), Ch.M. (Edin), PhD

Academic Editor

PLOS One

Journal Requirements:

Reviewers' comments:

Reviewer's Responses to Questions

**Comments to the Author**

1. If the authors have adequately addressed your comments raised in a previous round of review and you feel that this manuscript is now acceptable for publication, you may indicate that here to bypass the “Comments to the Author” section, enter your conflict of interest statement in the “Confidential to Editor” section, and submit your "Accept" recommendation.

Reviewer #1: All comments have been addressed

Reviewer #2: (No Response)

Reviewer #3: (No Response)

2. Is the manuscript technically sound, and do the data support the conclusions?

Reviewer #1: Yes

Reviewer #2: Partly

Reviewer #3: (No Response)

3. Has the statistical analysis been performed appropriately and rigorously? 

Reviewer #1: Yes

Reviewer #2: Yes

Reviewer #3: (No Response)

4. Have the authors made all data underlying the findings in their manuscript fully available?

Reviewer #1: Yes

Reviewer #2: Yes

Reviewer #3: (No Response)

5. Is the manuscript presented in an intelligible fashion and written in standard English?

Reviewer #1: Yes

Reviewer #2: No

Reviewer #3: (No Response)

6. Review Comments to the Author

Reviewer #1: I reviewed the revised manuscript and found that the authors have adequately addressed the comments and suggestions raised during the previous round of review.

Reviewer #2: This revised cost–utility analysis is substantially improved, with clearer comparator definition and better alignment between base-case and sensitivity analyses. The modeling framework (decision tree + Markov) is appropriate; however, key structural limitations remain. The “targeted screening” arm is not a true risk-stratified strategy but a proportional cost-reduction scenario without differential prevalence or test performance, limiting interpretability and external validity. This should be reframed more explicitly in Methods, not only Discussion. Absolute QALY magnitudes remain inflated at the cohort level and, while clarified, still risk misinterpretation; presenting per-patient QALYs alongside cohort totals would improve transparency. The small event count (3 MG, 1 thyroid) introduces instability in prevalence-driven ICER estimates, and probabilistic sensitivity analysis cannot fully compensate for structural uncertainty. Exclusion of implementation costs biases toward screening and should be emphasized as a directional limitation.

Clarify targeted strategy definition in Methods; report per-patient QALYs; explicitly emphasize structural uncertainty and small-event limitations; tighten language.

Reviewer #3: (No Response)

7. PLOS authors have the option to publish the peer review history of their article (what does this mean?). If published, this will include your full peer review and any attached files.

Reviewer #1: **Yes:** Rashmi Paudel

Reviewer #2: **Yes:** Robert J. Chen, MD

Reviewer #3: **Yes:** Luca Bertolaccini

---

## [Author Response · Author response to Decision Letter 3]

11 May 2026

Reviewers' comments:

Reviewer's Responses to Questions

Comments to the Author

1. If the authors have adequately addressed your comments raised in a previous round of review and you feel that this manuscript is now acceptable for publication, you may indicate that here to bypass the “Comments to the Author” section, enter your conflict of interest statement in the “Confidential to Editor” section, and submit your "Accept" recommendation.

Reviewer #1: All comments have been addressed

Reviewer #2: (No Response)

Reviewer #3: (No Response)

2. Is the manuscript technically sound, and do the data support the conclusions?

Reviewer #1: Yes

Reviewer #2: Partly

Reviewer #3: (No Response)

3. Has the statistical analysis been performed appropriately and rigorously?

Reviewer #1: Yes

Reviewer #2: Yes

Reviewer #3: (No Response)

4. Have the authors made all data underlying the findings in their manuscript fully available?

Reviewer #1: Yes

Reviewer #2: Yes

Reviewer #3: (No Response)

5. Is the manuscript presented in an intelligible fashion and written in standard English?

Reviewer #1: Yes

Reviewer #2: No

Reviewer #3: (No Response)

6. Review Comments to the Author

Reviewer #1: I reviewed the revised manuscript and found that the authors have adequately addressed the comments and suggestions raised during the previous round of review.

Reviewer #2: This revised cost–utility analysis is substantially improved, with clearer comparator definition and better alignment between base-case and sensitivity analyses. The modeling framework (decision tree + Markov) is appropriate; however, key structural limitations remain. The “targeted screening” arm is not a true risk-stratified strategy but a proportional cost-reduction scenario without differential prevalence or test performance, limiting interpretability and external validity. This should be reframed more explicitly in Methods, not only Discussion. Absolute QALY magnitudes remain inflated at the cohort level and, while clarified, still risk misinterpretation; presenting per-patient QALYs alongside cohort totals would improve transparency. The small event count (3 MG, 1 thyroid) introduces instability in prevalence-driven ICER estimates, and probabilistic sensitivity analysis cannot fully compensate for structural uncertainty. Exclusion of implementation costs biases toward screening and should be emphasized as a directional limitation.

Clarify targeted strategy definition in Methods; report per-patient QALYs; explicitly emphasize structural uncertainty and small-event limitations; tighten language.

Response:

We sincerely thank the reviewer for the thoughtful feedback and for recognizing the substantial improvements made in the revised manuscript.

In response to these final comments, we made several targeted revisions to further improve clarity and transparency.

First, we strengthened the description of the targeted screening strategy in the Methods section by explicitly stating that this approach represents a proportional reduction in testing volume rather than a formally risk-enriched screening strategy (on page 4, lines 117-119). This clarification was previously emphasized in the Discussion and has now been incorporated earlier in the manuscript.

Second, to improve transparency regarding QALY interpretation, we now report per-patient QALY estimates alongside cohort-level totals in the Results section (on page 8, lines 253-256 and page 9, lines 267-268 and table 2 legends). This helps avoid misinterpretation of large absolute cohort QALY values.

Third, we further strengthened the limitations section by explicitly stating that probabilistic sensitivity analysis addresses parameter uncertainty but cannot fully resolve structural uncertainty related to model assumptions and the small number of observed disease events (on page 15, lines 489-491).

Fourth, we revised the limitations section to more explicitly state that exclusion of implementation costs likely biases the model in favor of screening and may underestimate the full economic burden of universal implementation (on page 15, lines 497-499).

Finally, we made minor language refinements throughout the manuscript for clarity and corrected minor wording inconsistencies (on page 13, lines 399-401).

We appreciate the reviewer’s thoughtful comments, which have helped further strengthen the manuscript.

Reviewer #3: (No Response)

7. PLOS authors have the option to publish the peer review history of their article (what does this mean?). If published, this will include your full peer review and any attached files.

Do you want your identity to be public for this peer review? For information about this choice, including consent withdrawal, please see our Privacy Policy.

Reviewer #1: Yes: Rashmi Paudel

Reviewer #2: Yes: Robert J. Chen, MD

Reviewer #3: Yes: Luca Bertolaccini

---

## [Editor Report · Decision Letter 3]

12 May 2026

A Preliminary Cost-Utility Analysis of Routine Myasthenia Gravis and Thyroid Dysfunction Screening in Acquired Comitant Esotropia

PONE-D-25-52845R3

Dear Dr. Srimanan,

We’re pleased to inform you that your manuscript has been judged scientifically suitable for publication and will be formally accepted for publication once it meets all outstanding technical requirements.

Kind regards,

Redoy Ranjan, MS (CV&TS), Ch.M. (Edin), PhD

Academic Editor

PLOS One
---

## [Editor Report · Acceptance letter]

PONE-D-25-52845R3

PLOS One

Dear Dr. Srimanan,

I'm pleased to inform you that your manuscript has been deemed suitable for publication in PLOS One. Congratulations! Your manuscript is now being handed over to our production team.

Kind regards,

on behalf of

Dr. Redoy Ranjan

Academic Editor

PLOS One